# The Planar Thirring Model with Kähler-Dirac Fermions

Simon Hands [†]

Department of Physics, Swansea University, Singleton Park, Swansea SA2 8PP, UK;
Simon.Hands@liverpool.ac.uk or s.j.hands@swansea.ac.uk
† Current address: Department of Mathematical Sciences, University of Liverpool, Liverpool L69 3BX, UK.

**Abstract:** Kähler's geometric approach in which relativistic fermion fields are treated as differential forms is applied in three spacetime dimensions. It is shown that the resulting continuum theory is invariant under global U(N)⊗U(N) field transformations and has a parity-invariant mass term, which are symmetries shared in common with staggered lattice fermions. The formalism is used to construct a version of the Thirring model with contact interactions between conserved Noether currents. Under reasonable assumptions about field rescaling after quantum corrections, a more general interaction term is derived, sharing the same symmetries but now including terms which entangle spin and taste degrees of freedom, which exactly coincides with the leading terms in the staggered lattice Thirring model in the long-wavelength limit. Finally, truncated versions of the theory are explored; it is found that excluding scalar and pseudoscalar components leads to a theory of six-component fermion fields describing particles with spin 1, with fermion and antifermion corresponding to states with definite circular polarisation. In the UV limit, only transverse states with just four non-vanishing components propagate. Implications for the description of dynamics at a strongly interacting renormalisation group fixed point are discussed.

**Keywords:** interacting fermions; field theories in dimensions other than four; staggered lattice fermions; renormalisation group fixed point





## 1. Introduction

This paper concerns relativistic fermions interacting strongly in three spacetime dimensions, in the context of a field theory known as the Thirring model with Lagrangian density

$$\mathcal{L} = \bar{\psi}_i(\slashed{\partial} + m)\psi_i + \frac{g^2}{2N}(\bar{\psi}_i\gamma_\mu\psi_i)^2. \tag{1}$$

Here, the fields $\psi_i, \bar{\psi}_i$ are reducible spinors, so the Dirac matrices $\gamma_\mu$ are $4 \times 4$. In the Euclidean metric, they obey $\gamma_\mu = \gamma_\mu^\dagger$ and $\{\gamma_\mu, \gamma_\nu\} = 2\delta_{\mu\nu}$. The index $i = 1, \ldots, N$ runs over $N$ distinct fermion species. The contact interaction between conserved fermion currents $\bar{\psi}\gamma_\mu\psi$ results in a repulsive force between fermions but attraction between fermions and antifermions. A question of interest is then whether in the massless limit $m \to 0$ a bilinear condensate $\langle\bar{\psi}\psi\rangle \neq 0$ forms as a result of strong interactions, leading to the dynamical generation of fermion mass.

It is natural to analyse bilinear condensation in terms of symmetry breaking. In three dimensions, there are two elements of the reducible Dirac algebra $\gamma_4$ and $\gamma_5$, which anticommute with the kinetic term of (1). Accordingly, $m \to 0$ (1) is invariant under the following field rotations:

$$\psi \mapsto e^{i\alpha_1}\psi; \; \bar{\psi} \mapsto \bar{\psi}e^{-i\alpha_1} \quad : \quad \psi \mapsto e^{\alpha_{45}\gamma_4\gamma_5}\psi; \; \bar{\psi} \mapsto \bar{\psi}e^{-\alpha_{45}\gamma_4\gamma_5}; \tag{2}$$

$$\psi \mapsto e^{i\alpha_4\gamma_4}\psi; \; \bar{\psi} \mapsto \bar{\psi}e^{i\alpha_4\gamma_4} \quad : \quad \psi \mapsto e^{i\alpha_5\gamma_5}\psi; \; \bar{\psi} \mapsto \bar{\psi}e^{i\alpha_5\gamma_5}. \tag{3}$$

Together, these rotations generate a U(2N) global symmetry, which can be broken either explicitly by $m \neq 0$ or spontaneously by $\langle\bar{\psi}\psi\rangle \neq 0$ to U(N)⊗U(N), when the rotations

(3) no longer leave the ground state invariant. Goldstone's theorem implies spontaneous symmetry breaking yields $2N^2$ massless bosons in the theory's spectrum.

It is suspected that symmetry breaking occurs for a sufficiently large interaction strength $g^2$ and a sufficiently small $N$; it is even possible that the resulting quantum critical point observed at $g_c^2(N)$ might be a UV-stable fixed point of the renormalisation group, implying that a continuum limit at this point is possible. The fixed-point theory is expected to display universal features of the strongly interacting dynamics characterised by the pattern of symmetry breaking. However, there are no small parameters to enable a systematic investigation of this phenomenon by analytic means. Determination of the critical exponents and even the critical flavour number $N_c$ below which symmetry breaking can occur are essentially non-perturbative problems.

A natural approach employs numerical simulations of lattice field theory (a recent review can be found in [1]). The most recent work uses a lattice fermion formulation which seeks to respect the U(2$N$) symmetry such as the SLAC derivative [2,3], or domain wall fermions [4,5]. However, there is also a substantial body of earlier simulations [6] employing the more primitive staggered formulation, in which fermion fields are represented by single-component Grassmann objects $\chi, \bar{\chi}$ located on the sites of a cubic lattice. As well as U($N$) flavour rotations, staggered fermions also enjoy a second U($N$) global symmetry protecting them from acquiring mass, of the form

$$\chi \mapsto e^{i\beta\varepsilon_x}\chi; \ \bar{\chi} \mapsto e^{i\beta\varepsilon_x}\bar{\chi}, \tag{4}$$

where $\varepsilon_x = (-1)^{x_1+x_2+x_3}$ is an alternating sign in effect partitioning the sites $x$ into distinct odd and even sublattices. This time, therefore, bilinear condensation drives a symmetry breaking U($N$)⊗U($N$) →U($N$), resulting in just $N^2$ Goldstones. For a strongly interacting system, therefore, we expect distinct fixed-point behaviour and, indeed, simulations of the staggered model [7] support a critical $N_c \approx 3.3$ that is significantly larger than that found for the U(2$N$)-symmetric variants [3–5]. Moreover, simulation studies of the minimal staggered model with $N = 1$. (it is very common in the literature to designate $N$ staggered flavours in 3$d$ as describing $N_f = 2N$ "continuum flavors" [8]). Ref. [9] find critical indices indistinguishable from those of the Gross–Neveu model having the same global symmetries [10], even though in the latter case symmetry breaking can be described analytically using a $1/N$ expansion.

Despite these apparent shortcomings, the staggered Thirring model does exhibit interesting behaviour; in particular, the critical exponents characterising the fixed-point are sensitive to the value of $N < N_c$ [7]. Could there exist a continuum-based description of the corresponding fixed-point theories? One question which needs addressing is the significance of $N$—in a weak-coupling long-wavelength limit it is natural to interpret staggered femions in terms of $N_f = 2N$ autonomous flavours [8], or in modern parlance, each staggered flavour describes two continuum "tastes". However, even in early staggered Thirring studies [6], the factorisation of interaction currents into distinct and mutually independent taste sectors was not manifest, and there is no reason a priori to require this in a strongly coupled setting. In what follows, we will refer to the difficulty in separating taste and spin components as "spin/taste entanglement". A related question is how to engineer the U($N$)⊗U($N$) symmetry in the continuum where we have no lattice partition to help recover (4).

This paper will answer such questions using a framework introduced into lattice field theory by Becher and Joos in 1982 [11], who found that a version of the Dirac equation rooted in concepts of differential geometry originally noted by Kähler [12] in 1962 is in fact the formal continuum limit of staggered lattice fermions. As set out in the next few sections, in the Kähler-Dirac approach fermions are not spinor fields but rather are complexes of $p$-forms, where $p = 0, 1, \ldots, d$, with $d$ the dimension of spacetime. This is a natural way to prepare for transcribing continuum fields to a lattice [13]; indeed, it even proves possible to formulate Kähler-Dirac fermions on simplicial lattices [14], thereby extending the staggered approach to the random geometries explored in dynamical triangulation

models of quantum gravity. Each $p$-form has $_dC_p$ components. In the four dimensional case analysed in [11] a fermion field has thus $1 + 4 + 6 + 4 + 1 = 16$ components, which are recast as four independent tastes of four-component spinor fields. For the case $d = 3$ to be developed in what follows, the corresponding field has eight components recast as two tastes of reducible four-component spinor. The algebraic details very closely mirror the assignment of spin/taste degrees of freedom to staggered lattice fermions in three spacetime dimensions originally set out by Burden and Burkitt [8].

The remainder is organised as follows. Section 2 is a brief but hopefully self-contained introduction to the differential geometry machinery required. Readers who are already expert will find our notations and conventions set out; those less familiar might also benefit from the helpful Appendix of [11], or a textbook such as [15]. Section 3 derives the equivalence between the free Kähler-Dirac equation, which with suitable notation assumes the same form in any dimension, and a continuum Dirac equation in three Euclidean dimensions describing two tastes of reducible spinor. The same framework is used in Section 4, following the introduction of a generalised scalar product between $p$ forms, to identify the fermion current that will be used in building the Thirring interaction term. Section 5 at last introduces the Thirring model action in the Kähler-Dirac language, and identifies both the U($N$)⊗U($N$) global symmetry and also an important parity symmetry shared in common with staggered lattice fermions. The Euclidean path integral is introduced permitting an explicit derivation of the Noether current associated with the symmetry corresponding to (4).

In Section 6, we begin to take the geometrical form of the theory more seriously by exploring the idea that in a suitably regularised interacting theory the renormalisation of the field components should depend on $p$: the Thirring interaction term is modified in order to accommodate this possibility, and it is shown that the resulting terms when recast in a spinor basis exhibit spin/taste entanglement and are in exact correspondence with the interaction derived from the staggered Thirring model [6] using the formalism of [8]. This demonstrates that the proposed $p$-dependent field rescaling is perfectly consistent with a properly regularised lattice model, and also that spin/taste entanglement is not a lattice artefact, but rather in fact a feature of an interacting continuum field theory. Finally, in Section 7, the idea is taken a step further with the exploration of truncated actions resulting from retaining just field components with two consecutive values of $p$. The most interesting case corresponds to keeping just $p = 1, 2$, resulting in a theory of six-component spin-one fermions, whose physical states are transverse, and for which fermion and antifermion are states of opposite polarisation. Section 8 summarises the paper's findings and speculates on the applicability of the exotic scenario of Section 7 to the physics of a putative renormalisation group fixed point at strong coupling.

## 2. Mathematical Preliminaries

The theory to be developed uses the language of differential forms in three-dimensional Euclidean spacetime. We will follow the presentation and notation of [11] closely. In this approach, all physical quantities are viewed as $p$ forms defined in some vector space $^p\Lambda$, with $p = 0, 1, \ldots, 3$. A suitable basis for $^p\Lambda$ is given by $dx^H = dx^{\mu_1} \wedge \ldots \wedge dx^{\mu_p}$ with the exterior product satisfying

$$\wedge : {}^p\Lambda \times {}^q\Lambda \mapsto {}^{p+q}\Lambda, \quad dx^H \wedge dx^K = \begin{cases} \rho_{H,K} dx^{H \cup K} & \text{if } H \cap K = \varnothing; \\ 0 & \text{otherwise,} \end{cases} \tag{5}$$

where the sign factor $\rho_{H,K} = (-1)^s$, with $s$ the number of pairs $\mu, \nu \in H \times K$ with $\mu > \nu$, and $\rho_{\varnothing,H} = \rho_{H,\varnothing} = 1$. With this in place, any function $\Phi$ can be expanded as follows:

$$\begin{aligned} \Phi(x) &= \varphi_\varnothing(x) + \varphi_\mu(x)dx^\mu + \frac{1}{2!}\varphi_{\mu\nu}(x)dx^\mu \wedge dx^\nu + \varphi_{123}(x)dx^1 \wedge dx^2 \wedge dx^3 \\ &\equiv \sum_H \varphi(x,H)dx^H. \end{aligned} \tag{6}$$

The convention is that repeated indices are summed over, and no special significance is attached to whether an index is super- or subscript. In dealing with quantities defined on the whole space $\Lambda = \bigoplus_{p=0}^{3} {}^{p}\Lambda$, it is convenient to define the main automorphism

$$\mathcal{A} : \Lambda \mapsto \Lambda, \quad \mathcal{A}\Phi = \sum_{H} (-1)^{p(H)} \varphi(x,H) dx^{H}, \tag{7}$$

and the main antiautomorphism

$$\mathcal{B} : \Lambda \mapsto \Lambda, \quad \mathcal{B}\Phi = \sum_{H} (-1)^{p(H)C_2} \varphi(x,H) dx^{H}. \tag{8}$$

where for $p \in \{0,\dots,3\}$ the combinatoric factor ${}_p C_2$ takes values $\{0,0,1,3\}$.

Three key operations are then:

- Exterior derivative
$$d : {}^{p}\Lambda \mapsto {}^{p+1}\Lambda, \quad d\Phi = dx^{\mu} \wedge \partial_{\mu}\Phi \tag{9}$$

  Since $\partial_{\mu}\partial_{\nu} = \partial_{\nu}\partial_{\mu}$ it immediately follows from (5) that $d^2 = 0$.
- Hodge Star
$$\star : {}^{p}\Lambda \mapsto {}^{3-p}\Lambda, \quad \star dx^{H} = \rho_{H,\mathcal{C}H} dx^{\mathcal{C}H} \tag{10}$$

  where $\mathcal{C}H$ is the complement of $H$. In odd-dimensional Euclidean spacetimes, $\star\star = 1$.
- Co-derivative
$$\delta : {}^{p}\Lambda \mapsto {}^{p-1}\Lambda, \quad \delta = \star \mathcal{B} d \star \mathcal{B} = \star d \star \mathcal{A}. \tag{11}$$

  and it immediately follows from $d^2 = 0$, $\star\star = 1$ that $\delta^2 = 0$. The co-derivative's sign depends in general on $p$, $d$ and the signature of the metric [15], which in (11) is captured by the use of the automorphisms (7) and (8). A convenient representation for its action is

$$\delta\Phi = -e^{\mu}\lrcorner\partial_{\mu}\Phi, \tag{12}$$

  where the contraction operator enabling differentiation with respect to a differential is defined by

$$e^{K}\lrcorner dx^{H} = \begin{cases} \rho_{K,H\backslash K} dx^{H\backslash K} & \text{if } K \subset H; \\ 0 & \text{otherwise.} \end{cases} \tag{13}$$

### 3. The Kähler-Dirac Equation

The starting point is the observation that $(d - \delta)^2 = -(d\delta + \delta d) = \partial_{\mu}\partial_{\mu} = \Delta$, the Laplacian operator. Hence, $d - \delta$ is in effect the square-root of the Laplacian, and therefore linear in momentum, while still local. It is thus a candidate for incorporating in a relativistic wave equation, as first written by Kähler [12]:

$$(d - \delta + m)\Phi = 0. \tag{14}$$

The Kähler-Dirac equation (KDE) takes the same form in any spacetime dimension. The scalar parameter $m$ is the fermion mass. Note that, since $d$ and $\delta$ implement $\Delta p = \pm 1$, the equation only makes sense if $\Phi \in \Lambda$, i.e., $\Phi$ admits an expansion of the form (6), with components $\varphi(x,H)$ having a mass dimension of 1 in three spacetime dimensions.

It is helpful to define the Clifford product between differential forms:

$$\vee : \Lambda \times \Lambda \mapsto \Lambda, \ \Phi \vee \Xi = \sum_{p} \frac{(-1)^{pC_2}}{p!} (\mathcal{A}^{p} e^{\mu_1}\lrcorner \dots e^{\mu_p}\lrcorner \Phi) \wedge (e^{\mu_1}\lrcorner \dots e^{\mu_p}\lrcorner \Xi), \tag{15}$$

with particular instances

$$dx^{\mu} \vee \Phi = dx^{\mu} \wedge \Phi + e^{\mu}\lrcorner\Phi; \quad \Phi \vee dx^{\mu} = \Phi \wedge dx^{\mu} - e^{\mu}\lrcorner\mathcal{A}\Phi. \tag{16}$$

It immediately follows from (9) and (12) that the KDE can be rewritten

$$(dx^\mu \vee \partial_\mu + m)\Phi = 0. \tag{17}$$

Now, the identity

$$dx^\mu \vee dx^\nu \vee + dx^\nu \vee dx^\mu \vee = 2\delta^{\mu\nu} \tag{18}$$

is strongly reminiscent of the defining relation $\{\gamma_\mu, \gamma_\nu\} = 2\delta_{\mu\nu}$ for Dirac matrices in Euclidean metric, and suggests the operation $dx^\mu\vee$ furnishes a representation of the Dirac algebra in the eight-dimensional space spanned by $dx^H$. The appropriate representation of the algebra in three spacetime dimensions was identified in [8] in a study of the staggered lattice fermion operator. It is the direct sum $\sigma_H \oplus \tau_H$ of two inequivalent irreducible two-dimensional representations generated by the Pauli matrices $\sigma_\mu$ ($\mu = 1, 2, 3$), and by $\tau_\mu = -\sigma_\mu$. The Pauli matrices have the property $\sigma_\mu^* = \sigma_\mu^T$, where * denotes a complex conjugation and $T$ the matrix transpose. Analysis proceeds by identifying a new basis

$$
\begin{aligned}
\Sigma \oplus T &= 1 + (\sigma_\mu^T \oplus \tau_\mu^T)dx^\mu + \frac{1}{2!}(\sigma_\mu^T \sigma_\nu^T \oplus \tau_\mu^T \tau_\nu^T)dx^\mu \wedge dx^\nu \\
&+ (\sigma_1^T \sigma_2^T \sigma_3^T \oplus \tau_1^T \tau_2^T \tau_3^T)dx^1 \wedge dx^2 \wedge dx^3 = \sum_H \mathcal{B}(\sigma \oplus \tau)_H^T dx^H
\end{aligned} \tag{19}
$$

The key result is now

$$dx^\alpha \vee (\Sigma \oplus T) = (\sigma^\alpha \oplus \tau^\alpha)^T (\Sigma \oplus T) = \sum_c (\sigma^\alpha \oplus \tau^\alpha)_{ac}^T (\Sigma \oplus T)_{cb}, \tag{20}$$

where Roman indices $a, b, c = 1, 2$. The derivation of (20) makes repeated use of $\{(\sigma \oplus \tau)_\mu, (\sigma \oplus \tau)_\nu\} = 2\delta_{\mu\nu} \otimes \mathbb{1}_{2\times2}$.

In order to express the KDE in the basis (19), we need the orthogonality relations

$$\sum_H \sigma_{ab}^H \sigma_{cd}^{H*} = \sum_H \tau_{ab}^H \tau_{cd}^{H*} = 4\delta_{ac}\delta_{bd}; \quad \sum_H \sigma_{ab}^H \tau_{cd}^{H*} = 0, \tag{21}$$

implying

$$dx^H = \frac{1}{4}(-1)^{pC_2}\text{tr}(\sigma_H \oplus \tau_H)^*(\Sigma \oplus T). \tag{22}$$

Using (6), we then define

$$\Phi(x) = \sum_H \varphi(x, H)dx^H = \sum_{a,b} u_a^b(x)\Sigma_a \oplus d_a^b(x)T_a \tag{23}$$

where we have introduced fields $u$, $d$ whose lower index $a = 1, 2$ will turn out to be associated with spinor degrees of freedom in the non-interacting case and whose upper index $b = 1, 2$ will be associated with taste. The field transformations between bases are then:

$$
\begin{aligned}
\varphi(x, H) &= \sum_{a,b} u_a^b(x)(\sigma_H^T)_{ab} \oplus d_a^b(x)(\tau_H^T)_{ab}; \tag{24}\\
(u_a^b \oplus d_a^b)(x) &= \frac{1}{4}\sum_H \varphi(x, H)(\sigma \oplus \tau)_{ab}^H. \tag{25}
\end{aligned}
$$

Combining the result (20) with the KDE equation (17) we deduce

$$(\gamma_\mu \partial_\mu + m)\psi^b(x) = 0, \tag{26}$$

i.e., the free Dirac equation for a two-taste four-component spinor field $\psi = u \oplus d$, with Euclidean Dirac matrices defined

$$\gamma_\mu = \begin{pmatrix} \sigma_\mu & \\ & \tau_\mu \end{pmatrix}. \tag{27}$$

We will refer to this familar form as the free KDE in the $\psi$-basis.

## 4. Interaction Current

In order to develop an interacting theory we will need a definition of a current in the Kähler-Dirac formalism. This requires the definition of a generalised scalar product $(,)_p : \Lambda \times \Lambda \mapsto^{3-p} \Lambda$ [11,12]. The two cases we will need have $p = 0$:

$$(\Phi, \Xi)_0 = (\mathcal{B}\Phi \vee \Xi) \wedge \varepsilon \tag{28}$$

and $p = 1$:

$$(\Phi, \Xi)_1 = e_\mu \lrcorner (dx^\mu \vee \Phi, \Xi)_0 = e_\mu \lrcorner [(dx^\mu \vee \Phi \vee \mathcal{B}\Xi) \wedge \varepsilon], \tag{29}$$

where $\varepsilon$ is the volume three-form $dx^1 \wedge dx^2 \wedge dx^3$. In components, these are expressed

$$(\Phi, \Xi)_0 = \left[ \varphi_\varnothing \xi_\varnothing + \varphi_\mu \xi_\mu + \frac{1}{2!} \varphi_{\mu\nu} \xi_{\mu\nu} + \varphi_{123} \xi_{123} \right] \varepsilon = \sum_H \varphi(x, H) \xi(x, H) \varepsilon; \tag{30}$$

and

$$(\Phi, \Xi)_1 = \frac{1}{2!} \left[ \varphi_\varnothing \xi_\alpha + \varphi_\alpha \xi_\varnothing + \varphi_\mu \xi_{\alpha\mu} + \varphi_{\alpha\mu} \xi_\mu + \frac{1}{2!} \left( \varphi_{\mu\nu} \xi_{\alpha\mu\nu} + \varphi_{\alpha\mu\nu} \xi_{\mu\nu} \right) \right] \epsilon_{\alpha\mu\nu} dx^\mu \wedge dx^\nu \tag{31}$$

The following Green's formula identity is useful [12]:

$$d(\Phi, \Xi)_1 = (\Phi, (d - \delta)\Xi)_0 + ((d - \delta)\Phi, \Xi)_0. \tag{32}$$

Next, define $\bar{\Phi} = \mathcal{A}\Phi^*$ as the solution of the adjoint KDE:

$$(d - \delta - m)\bar{\Phi} = 0. \tag{33}$$

A current one-form is then given by

$$j = j_\mu dx^\mu = - \star \frac{i}{4} (\bar{\Phi}, \Phi)_1. \tag{34}$$

Current conservation follows using (14), (32) and (33):

$$\delta j = - \star d \star j \quad = \quad \star \frac{i}{4} d(\bar{\Phi}, \Phi)_1 = \star \frac{i}{4} [(\bar{\Phi}, (d - \delta)\Phi)_0 + ((d - \delta)\bar{\Phi}, \Phi)_0] \tag{35}$$

$$= \quad - \star \frac{i}{4} (\bar{\Phi}, \Phi)_0 (m - m) = 0,$$

i.e., $\partial_\mu j_\mu = 0$. Now, use $(\Phi, \Xi)_p = (-1)^{pC_2} (\Xi, \Phi)_p$ and (20) to write

$$(\bar{\Phi}, \Phi)_1 = e_\mu \lrcorner (\bar{\Phi}, dx^\mu \vee \Phi)_0 = e_\mu \lrcorner (\bar{u}\Sigma^* \oplus \bar{d}T^*, (\sigma^\mu u)\Sigma \oplus (\tau^\mu d)T)_0. \tag{36}$$

In the $\psi$-basis the current one-form thus reads

$$j = i \sum_b \bar{\psi}^b(x) \gamma_\mu \psi^b(x) dx^\mu. \tag{37}$$

## 5. Action and Symmetries

Now, we have enough equipment to define the action and hence the Euclidean path integral. The action for free fields is

$$S_0 = \frac{1}{4} \int (\bar{\Phi}, (d - \delta + m)\Phi)_0 = \sum_{b=1,2} \int \bar{\psi}^b (\gamma_\mu \partial_\mu + m)\psi^b \varepsilon. \tag{38}$$

For the Thirring model, this is supplemented by a contact interaction of the form $-\frac{g^2}{2} j_\mu j_\mu$, where the normalisation of the coupling strength, which has mass dimension-1, is somewhat conventional but has been chosen to be consistent with [6]. In form notation, this reads

$$-\frac{g^2}{2} \int (j, j)_0 = -\frac{g^2}{2} \int (j \vee j) \wedge \varepsilon = \frac{g^2}{32} \int (\star(\bar{\Phi}, \Phi)_1 \vee \star(\bar{\Phi}, \Phi)_1) \wedge \varepsilon. \tag{39}$$

Using $(\star\Phi, \star\Xi)_0 = (\Phi, \Xi)_0$, we arrive at the Thirring model action

$$S = \int \frac{1}{4}(\bar{\Phi}, (d - \delta + m)\Phi)_0 + \frac{g^2}{32}((\bar{\Phi}, \Phi)_1 \vee (\bar{\Phi}, \Phi)_1) \wedge \varepsilon \tag{40}$$

$$= \int \left[ \bar{\psi}^b (\gamma_\mu \partial_\mu + m)\psi^b + \frac{g^2}{2}(\bar{\psi}^b \gamma_\mu \psi^b)(\bar{\psi}^c \gamma_\mu \psi^c) \right] \varepsilon. \tag{41}$$

We note in passing that four-fermi interactions in models constructed from Kähler-Dirac fermions have also been investigated in four dimensions [16].

As a consequence of its construction from $(\bar{\Phi}, \Phi)$ bilinears the action (40) has two manifest global symmetries. First:

$$\Phi \mapsto e^{i\theta}\Phi; \quad \bar{\Phi} \mapsto e^{-i\theta}\bar{\Phi}. \tag{42}$$

This symmetry correponds to the conservation of fermion charge, and the corresponding Noether current is given by (34). Second, in the limit $m \to 0$:

$$\Phi \mapsto e^{i\omega\mathcal{A}}\Phi; \quad \bar{\Phi} \mapsto e^{i\omega\mathcal{A}}\bar{\Phi}, \tag{43}$$

which follows because $d, \delta$ both yield $\Delta p = \pm 1$ and by inspection of the component expansion of $(\bar{\Phi}, \Phi)_1$ (31). This is analogous to the chiral symmetry protecting fermions from additive mass renormalisation in $d = 4$. The corresponding Noether current is

$$j_\mathcal{A} = -\star \frac{i}{4}(\bar{\Phi}, \mathcal{A}\Phi)_1. \tag{44}$$

In order to translate to the $\psi$-basis, observe that the action of $\mathcal{A}$ in effect exchanges $\sigma_H$ and $\tau_H$ in (25). It then follows straightforwardly that

$$j_{\mathcal{A}\mu} = i\bar{\psi}^b \gamma_\mu \gamma_5 \psi^b \tag{45}$$

where we introduce two new hermitian $\gamma$-matrices obeying $\{\gamma_4, \gamma_\mu\} = \{\gamma_5, \gamma_\mu\} = \{\gamma_4, \gamma_5\} = 0$:

$$\gamma_4 = \begin{pmatrix} & -i\mathbb{1} \\ i\mathbb{1} & \end{pmatrix}; \quad \gamma_5 = \begin{pmatrix} & \mathbb{1} \\ \mathbb{1} & \end{pmatrix}. \tag{46}$$

From here it is straightforward to extend the model by introducing $N$ Kähler-Dirac fermion flavours $\Phi^i$, $i = 1, \ldots, N$. The flavour index $i$ is distinct from the indices $b, c = 1, 2$ in (41), which run over taste degrees of freedom. The two U(1) rotation symmetries (42) and (43) are trivially extended to U($N$)$\otimes$U$_\mathcal{A}$($N$), broken to U($N$) either explicitly by $m \neq 0$, or spontaneously by dynamical generation of a non-vanishing condensate $\langle (\bar{\Phi}^i, \Phi^i)_0 \rangle$.

Finally, consider discrete parity inversion. In odd spacetime dimensions, this is conveniently represented by inversion of all spacetime axes: $x_\mu \mapsto -x_\mu, \partial_\mu \mapsto -\partial_\mu$. The action (40) and (41) is invariant provided

$$\Phi(x) \mapsto \mathcal{A}\Phi(-x); \ \bar{\Phi}(x) \mapsto \mathcal{A}\bar{\Phi}(-x) \Rightarrow \psi(x) \mapsto \gamma_5\psi(-x); \ \bar{\psi}(x) \mapsto \bar{\psi}(-x)\gamma_5. \quad (47)$$

Note that the Noether currents (34) and (45), along with all bilinears of the form $(\Phi, \Xi)_1$, are parity-odd.

The Euclidean path integral is defined by

$$\mathcal{Z} = \int D\Phi D\bar{\Phi} \exp(-S[\Phi, \bar{\Phi}]) \quad (48)$$

where $\Phi, \bar{\Phi}$ are now Grassmann-valued and $\bar{\Phi}$ is considered independent of $\Phi$. We illustrate its use via a derivation of the Ward Identity for the divergence of the current $j_{\mathcal{A}}$; for simplicity, we consider only the free action (38). Consider the impact of the field transformation (43) where $\omega(x)$ is infinitesimal but now spacetime-dependent.

$$
\begin{aligned}
S_0 \mapsto S_0' &= \frac{1}{4} \int \left( e^{i\omega(x)\mathcal{A}}\bar{\Phi}, (d - \delta + m)e^{i\omega(x)\mathcal{A}}\Phi \right)_0 \\
&= \frac{1}{4} \int \left( e^{i\omega\mathcal{A}}\bar{\Phi}, dx^\mu \vee \partial_\mu e^{i\omega\mathcal{A}}\Phi + m e^{i\omega\mathcal{A}}\Phi \right)_0 \\
&= S_0 + \frac{1}{4} \int i\partial_\mu\omega(\bar{\Phi}, dx^\mu \vee \mathcal{A}\Phi)_0 + im\omega((\bar{\Phi}, \mathcal{A}\Phi)_0 + (\mathcal{A}\bar{\Phi}, \Phi)_0).
\end{aligned}
\quad (49)
$$

Now, use (29) together with $(\Phi, \Xi)_p = (-1)^{pC_2}(\Xi, \Phi)_p$ and the definition (44) to write

$$
\begin{aligned}
S_0' - S_0 &= -\int (\partial_\mu\omega)dx^\mu \wedge \star j_{\mathcal{A}} + i\frac{m}{2} \int \omega(\bar{\Phi}, \mathcal{A}\Phi)_0 \\
&= \int \omega \left( d \star j_{\mathcal{A}} + i\frac{m}{2}(\bar{\Phi}, \mathcal{A}\Phi)_0 \right),
\end{aligned}
\quad (50)
$$

where in the second step we have integrated the first term by parts. Since the path-integral measure $D\Phi D\bar{\Phi} = \prod_{x,H} d\varphi(x, H)d\bar{\varphi}(x, H)$ is formally invariant under the field transformation (in fact, this is only the case on a spacetime manifold with vanishing Euler characteristic [17]), the change in variables has no impact on the path integral, and we conclude

$$\left\langle \int \omega \left[ -\star \delta j_{\mathcal{A}} + i\frac{m}{2}(\bar{\Phi}, \mathcal{A}\Phi)_0 \right] \right\rangle = 0. \quad (51)$$

Since (51) holds for any $\omega(x)$, we conclude the expectation value of the three-form in square brackets is identically zero, which is the Ward Identity. In the $\psi$-basis, it has the familiar form

$$\langle \partial_\mu\bar{\psi}\gamma_\mu\gamma_5\psi - 2m\bar{\psi}\gamma_5\psi \rangle = 0. \quad (52)$$

## 6. Impact of Quantum Corrections

Our treatment up to this point has been either classical or formal. In any application to a genuine interacting quantum field theory, it is inevitable that the theory will need to be regularised somehow in order to control the calculation of quantum corrections. As a concrete example, we have already discussed the close parallels between the KDE continuum formalism and staggered lattice fermions, and will assume without further discussion that the proof of [11] that the KDE is the formal continuum limit of staggered fermions continues to apply in three dimensions.

Regularisation is essentially some kind of truncation of the degrees of freedom present in the classical field theory and inevitably violates some of the symmetries of the classical theory. In many cases, this leads to the requirement of renormalisation of both the fields and

the coupling parameters in the theory, which depends on some physical scale. Consider the Thirring action in the $\psi$-basis (41), where the rotations (42) and (43) take the form

$$\psi \mapsto e^{i\theta}\psi; \quad \bar\psi \mapsto \bar\psi e^{-i\theta}: \quad \psi \mapsto e^{i\theta\gamma_5}\psi; \quad \bar\psi \mapsto \bar\psi e^{i\theta\gamma_5} \tag{53}$$

Equation (41) also looks to be invariant under a U(2) rotation among the tastes indexed by $b, c$. Beyond that, in the limit $m \to 0$ there are apparently additional symmetries corresponding to all the rotations given in (2) and (3), which together with taste rotations would generate a U(4N) global symmetry broken to U(2N)⊗U(2N) by a mass $m \neq 0$. Our viewpoint is that this symmetry is not fundamental and can only be recovered in certain limits, such as long wavelength or weak coupling.

We will proceed on the assumption that the geometric description employed in the KDE is more natural, so that after quantum corrections the field expansion of Equation (6) is modified:

$$\Phi_r(x) = \sum_H Z_{p(H)} \varphi(x, H) dx^H. \tag{54}$$

Here, a renormalised field $\Phi_r$ is defined in terms of bare components $\varphi(x, H)$ via wavefunction renormalisation constants $Z_p$ which depend on the interaction strength, the renormalisation scale and, crucially in this context, on the form degree $p$. This correction is covariant, in the sense that $Z_p$ is insensitive to rotations acting on the spacetime indices specific to $\varphi(x, H)$, and the key symmetries (42) and (43) continue to be respected by $\Phi_r$ even with $Z_p \neq 1$.

The form of (54) motivates a more general exploration of possible interaction currrents. In $d = 3$ the space of bilinear currents consistent with the four renormalisation constants $Z_p$ is spanned by $\star(\bar\Phi, \Phi)_1$, $\star(\bar\Phi, \mathcal{A}\Phi)_1$, $\star(\mathcal{B}\bar\Phi, \Phi)_1$ and $\star(\mathcal{B}\bar\Phi, \mathcal{A}\Phi)_1$. Transcription to the $\psi$-basis for the first two of these is given in (34) and (45), and e.g.,

$$
\begin{aligned}
(\mathcal{B}\bar\Phi, \Phi)_1 &= e_\mu \lrcorner (\mathcal{B}\bar\Phi, dx^\mu \vee \Phi)_0 \\
&= e_\mu \lrcorner \varepsilon \sum_H (-1)^{pC_2} \left[ \bar u \sigma_H \oplus \bar d \tau_H \right] \left[ (\sigma_\mu u)\sigma_H^* \oplus (\tau_\mu d)\tau_H^* \right].
\end{aligned}
\tag{55}
$$

Now, observe the following identities for the components of $\sigma_H$:

$$
\begin{aligned}
\sum_\rho \sigma_\rho \mathbb{1} \sigma_\rho &= 3; \quad \sum_\rho \sigma_\rho \sigma_1 \sigma_2 \sigma_3 \sigma_\rho = 3\sigma_1 \sigma_2 \sigma_3; \\
\sum_\rho \sigma_\rho \sigma_\mu \sigma_\rho &= -\sigma_\mu; \quad \sum_\rho \sigma_\rho \sigma_\mu \sigma_\nu \sigma_\rho = -\sigma_\mu \sigma_\nu.
\end{aligned}
\tag{56}
$$

Recalling $\tau_H = (-1)^p \sigma_H$, we deduce a particularly convenient combination:

$$
\begin{aligned}
&- \star (\bar\Phi + 2\mathcal{B}\bar\Phi, \Phi)_1 \\
&= dx^\mu \sum_H \sum_\rho \left[ \bar u \sigma_H \oplus \bar d \tau_H \right] \left[ (\tau_\rho \tau_\mu d \tau_\rho^*)_a^b (\sigma_H^*)_{ab} \oplus (\sigma_\rho \sigma_\mu u \sigma_\rho^*)_a^b (\tau_H^*)_{ab} \right] \\
&= 4 \sum_\rho (\bar u, \bar d) \begin{pmatrix} & -\sigma_\rho \sigma_\mu \otimes \sigma_\rho^* \\ +\sigma_\rho \sigma_\mu \otimes \sigma_\rho^* & \end{pmatrix} \begin{pmatrix} u \\ d \end{pmatrix} dx^\mu \\
&= 4 \sum_\rho \bar\psi (i\gamma_4 \gamma_\rho \gamma_\mu \otimes \tau_\rho^*)\psi dx^\mu.
\end{aligned}
\tag{57}
$$

Here, the second component of the tensor product is a $2 \times 2$ matrix acting on taste indices. Similarly,

$$- \star (\bar\Phi + 2\mathcal{B}\bar\Phi, \mathcal{A}\Phi)_1 = -4 \sum_\rho \bar\psi (i\gamma_4 \gamma_5 \gamma_\rho \gamma_\mu \otimes \tau_\rho^*)\psi dx^\mu. \tag{58}$$

In either case, what emerges is an interaction current which although parity-odd and respecting the U(1)⊗U$_\mathcal{A}$(1) symmetries (42) and (43) no longer treats fermion tastes

as independent degrees of freedom but rather entangles taste and spacetime rotations, contrary to what is expected for particle flavour degrees of freedom. Remarkably, the currents $\star(\bar{\Phi}, \Phi_1)$, $\star(\bar{\Phi}, \mathcal{A}\Phi)_1$, $\star([1 + 2\mathcal{B}]\bar{\Phi}, \Phi)_1$ and $\star([1 + 2\mathcal{B}]\bar{\Phi}, \mathcal{A}\Phi)_1$ all feature in equal weight contact interactions in the Thirring model formulated with staggered fermions on a $3d$ cubic lattice as derived in a basis with explicit spinor and taste indices using the formalism of [8], and given in Equation (2.12) of [6]. In view of the equivalence [11] between Kähler-Dirac fermions and the formal continuum limit of staggered lattice fermions, this result should not be surprising.

It is now clear that these interactions survive the long-wavelength $a \to 0$ limit, where the lattice spacing $a$ furnishes an explicit UV cutoff. Other terms entangling spinor and taste degrees of freedom, which formally vanish as $O(a)$, are also present in the lattice formulation [6]. The current analysis demonstrates that spin/taste entanglement is not a lattice artifact, but is rooted in a continuum action of the form (40) with $U(N) \otimes U_{\mathcal{A}}(N)$ symmetry. However, it is significant that such terms also emerge from a well-defined regularisation capable of application to strongly interacting dynamics.

## 7. Reduced Kähler-Dirac Fermions

Kähler-Dirac fermions offer a new language with which to discuss relativistic fermion dynamics. To quote Becher and Joos, "This differential geometric description of fermions might be a basis for the construction of different kinds of field theoretic model" [11]. Once the differential geometric scaffolding has been removed, what kind of stories will we be able to tell? With motivation coming from a desire to understand novel structures at strongly interacting fixed points, in this section we will hazard some speculations.

Let us start by expressing the free action (38) in the $\varphi$-basis (6), with Lagrangian density

$$
\begin{aligned}
\star \frac{1}{4}(\bar{\Phi}, (d-\delta)\Phi)_0 \quad = \quad & \frac{1}{4}\Big[\bar{\varphi}_{\varnothing}\partial_\mu \varphi_\mu + \bar{\varphi}_\mu(\partial_\mu \varphi_{\varnothing} + \partial_\nu \varphi_{\nu\mu}) \\
+ \quad & \bar{\varphi}_{\mu\nu}\left(\partial_\mu \varphi_\nu + \frac{1}{2!}\partial_\lambda \varphi_{\lambda\mu\nu}\right) + \frac{1}{2!}\bar{\varphi}_{\mu\nu\lambda}\partial_\mu \varphi_{\nu\lambda}\Big].
\end{aligned}
\tag{59}
$$

Each term in (59) is separately invariant under $U(1) \otimes U_{\mathcal{A}}(1)$ and parity (47). Now consider a reduced action containing just a subset of the $p$-form fields $\varphi(x, H)$. The motivation comes from Equation (54), where we envisage a partition of $\{0, 1, 2, 3\}$ into sets $P, Q$ with $Z_{p \in P} \gg Z_{p \in Q} \approx 0$ arising, say, as a consequence of large anomalous scaling dimensions at a renormalisation group fixed point. Clearly only cases retaining consecutive values of $p$ will result in propagating states. We consider two examples.

### 7.1. $P = \{0, 1\}$

If we truncate the field content to just $p = 0, 1$, there are four components $\phi \equiv (\varphi_{\varnothing}, \varphi_\mu)^T$ to keep track of. The Lagrangian density is

$$
\mathcal{L}_{01} = \frac{1}{4}[\bar{\varphi}_{\varnothing}\partial_\mu \varphi_\mu + \bar{\varphi}_\mu \partial_\mu \varphi_{\varnothing} + m\bar{\varphi}_{\varnothing}\varphi_{\varnothing} + m\bar{\varphi}_\mu \varphi_\mu] \equiv \frac{1}{4}\bar{\phi}M\phi.
\tag{60}
$$

The $4 \times 4$ matrix $M$ has

$$
\det M = m^2(m^2 - \Delta),
\tag{61}
$$

which therefore vanishes identically for massless fermions. The propagator $M^{-1}$ has components

$$
\begin{aligned}
\langle \varphi_{\varnothing}\bar{\varphi}_{\varnothing}\rangle \quad = \quad & \frac{m}{m^2 - \Delta}; \quad \langle \varphi_\mu \bar{\varphi}_\nu \rangle = \frac{m\delta_{\mu\nu}}{m^2 - \Delta} - \frac{\mathcal{P}_{\mu\nu}\Delta}{m(m^2 - \Delta)}; \\
\langle \varphi_{\varnothing}\bar{\varphi}_\mu \rangle \quad = \quad & \langle \varphi_\mu \bar{\varphi}_{\varnothing}\rangle = -\frac{\partial_\mu}{m^2 - \Delta},
\end{aligned}
\tag{62}
$$

where the transverse projector

$$\mathcal{P}_{\mu\nu} = \delta_{\mu\nu} - \frac{\partial_\mu \partial_\nu}{\Delta} \tag{63}$$

such that $\mathcal{P}_{\mu\nu}\partial_\mu = 0$, $\mathcal{P}_{\mu\lambda}\mathcal{P}_{\lambda\nu} = \mathcal{P}_{\mu\nu}$, and $\text{tr}\mathcal{P} = 2$. In momentum space, all components manifest a particle pole at $k^2 = -m^2$, but asymptotically scale differently: $\oslash\oslash \sim k^{-2}$; $\mu\nu \sim k^{-2} + k^0$; $\oslash\mu \sim k^{-1}$. We conclude that $\mathcal{L}_{01}$ describes particles of mass $m$, and that the resulting effective theory is well-behaved in the IR regime $k^2 \lesssim m^2$. The singular part in the $m \to 0$ limit has a vanishing longitudinal component.

Note also that following a field redefinition $\varphi_{\mu\nu} = \epsilon_{\mu\nu\lambda}\xi_\lambda$; $\varphi_{\mu\nu\lambda} = \epsilon_{\mu\nu\lambda}\xi_\oslash$, the action $\mathcal{L}_{23}[\xi_{\oslash,\mu}, \bar\xi_{\oslash,\mu}]$ yields an action identical in form to (60), the only difference being that in contrast to (60), the field $\xi_\oslash$ has negative intrinsic parity and $\xi_\mu$ is positive.

*7.2. $P = \{1, 2\}$*

In this case, there are six field components $\varphi_\mu$, $\varphi_{\mu\nu}$ with Lagrangian density

$$\mathcal{L}_{12} = \frac{1}{4}\left[\bar\varphi_\mu \partial_\nu \varphi_{\nu\mu} + \bar\varphi_{\mu\nu}\partial_\mu \varphi_\nu + m\bar\varphi_\mu \varphi_\mu + \frac{m}{2!}\bar\varphi_{\mu\nu}\varphi_{\mu\nu}\right]. \tag{64}$$

After a field redefinition

$$\chi_\mu = \varphi_\mu; \quad \bar\chi_\mu = \frac{1}{2!}\epsilon_{\mu\nu\lambda}\bar\varphi_{\nu\lambda}; \quad \bar\xi_\mu = \bar\varphi_\mu; \quad \xi_\mu = \frac{1}{2!}\epsilon_{\mu\nu\lambda}\varphi_{\nu\lambda}, \tag{65}$$

$\mathcal{L}_{12}$ can be rewritten

$$\mathcal{L}_{12} = \frac{1}{4}\left[\epsilon_{\mu\nu\lambda}\left[\bar\chi_\mu\partial_\nu\chi_\lambda - \bar\xi_\mu\partial_\nu\xi_\lambda\right] + m(\bar\chi_\mu\xi_\mu + \bar\xi_\mu\chi_\mu)\right] \tag{66}$$

$$= \frac{1}{4}(\bar\chi, \bar\xi)\begin{pmatrix} \partial_\mu\lambda^\mu & m\mathbb{1}_{3\times3} \\ m\mathbb{1}_{3\times3} & -\partial_\mu\lambda^\mu \end{pmatrix}\begin{pmatrix} \chi \\ \xi \end{pmatrix} \equiv \frac{1}{4}\bar{Y}MY, \tag{67}$$

where the six component fermion fields $Y = \chi \oplus \xi$. The individual kinetic terms for the three-component objects $\chi, \xi$ in (66) superficially resemble the Chern–Simons action for gauge boson fields in $d = 3$, and in (67) the $3 \times 3$ matrices $\lambda^\mu$, $\mu = 1, \ldots, 3$ are antihermitian generators of the spin-one representation of SU(2), i.e., obeying $[\lambda_\mu, \lambda_\nu] = -\epsilon_{\mu\nu\rho}\lambda_\rho$, and each with eigenvalues $i\lambda = 0, \pm1$. Fields respond to rotations in the $\rho\sigma$ plane via $(\chi, \xi)_\mu^T \mapsto (\Lambda_1)_{\mu\nu}(\chi, \xi)_\nu^T$ with

$$\Lambda_1(\theta_{\rho\sigma}) = \exp\left(-\frac{\theta_{\rho\sigma}}{2!}[\lambda^\rho, \lambda^\sigma]\right). \tag{68}$$

Unlike Dirac matrices the $\lambda^\mu$ do not obey a Clifford algebra, so the $6 \times 6$ matrix $M$ in (67) is less straightforward to invert than a conventional Dirac operator. We start by checking its determinant, introducing the notation $\partial_\mu\lambda^\mu \equiv \partial \cdot \lambda$:

$$\det M = -\det\left(m^2\mathbb{1}_{3\times3} + (\partial\cdot\lambda)^2\right) \tag{69}$$

$$= -\exp\left[\text{tr}(\ln m^2) + \text{tr}\left(+\frac{(\partial\cdot\lambda)^2}{m^2} - \frac{(\partial\cdot\lambda)^4}{2m^4} + \frac{(\partial\cdot\lambda)^6}{3m^6} - \cdots\right)\right]$$

Now, use $(\partial\cdot\lambda)^2 = -\Delta\mathcal{P}$ and $\text{tr}\mathcal{P} = 2$ to write

$$\det M = -\exp\left(3\ln m^2 + 2\ln(1 - \frac{\Delta}{m^2})\right) = -m^2(m^2 - \Delta)^2. \tag{70}$$

Again, the determinant vanishes if $m = 0$. The propagator exists for $m \neq 0$ and is given by

$$\langle Y\bar{Y}\rangle = \frac{1}{m^2 - \Delta}\begin{pmatrix} \partial\cdot\lambda & m - m^{-1}\partial_\mu\partial_\nu \\ m - m^{-1}\partial_\mu\partial_\nu & -\partial\cdot\lambda \end{pmatrix}. \tag{71}$$

Also note that

$$\frac{-1}{\Delta(m^2-\Delta)}\begin{bmatrix}(\partial\cdot\lambda)^3 & m(\partial\cdot\lambda)^2 \\ m(\partial\cdot\lambda)^2 & -(\partial\cdot\lambda)^3\end{bmatrix}M = \frac{1}{m^2-\Delta}\begin{bmatrix}\partial\cdot\lambda & m \\ m & -\partial\cdot\lambda\end{bmatrix}\mathcal{P}M$$

$$= \mathcal{P}_{\mu\nu}\otimes\mathbb{1}_{2\times2} \tag{72}$$

The fact that $M$ in the massless limit is invertible when acting on a transverse subspace is reminiscent of gauge theories, where the same issue occurs due to the redundancy of the field description as a consequence of an underlying invariance of the action under local gauge transformations of the form $A_\mu \mapsto A_\mu + \partial_\mu\Lambda$, with $A_\mu$ the vector potential. We can trace this to the invariance of (66), after integration by parts, under

$$\chi_\mu \mapsto \chi_\mu + \partial_\mu\vartheta_\chi; \quad \bar{\chi}_\mu \mapsto \bar{\chi}_\mu + \partial_\mu\vartheta_{\bar{\chi}}; \quad \xi_\mu \mapsto \xi_\mu + \partial_\mu\vartheta_\xi; \quad \bar{\xi}_\mu \mapsto \bar{\xi}_\mu + \partial_\mu\vartheta_{\bar{\xi}}. \tag{73}$$

Here, the $\vartheta(x)$ are Grassmann-valued fields and the subscripts emphasise that independent shifts are applied to each fermi field. For this reason, the mass term is not in generally invariant under (73), consistent with the fact that $M$ is invertible once $m \neq 0$. Further note that the textbook solution to defining a gauge-field propagator, namely to fix a gauge by adding a covariant term of the form $\zeta^{-1}(\partial_\mu A_\mu)^2$ to the action, would in this case yield terms of the form, e.g., $\sim(\epsilon_{\mu\nu\lambda}\partial_\mu\bar{\varphi}_{\nu\lambda})(\partial_\rho\varphi_\rho)$, consistent with U(1)$_A$ but violating parity. Rather, it makes more sense to regard the term $m(\bar{\chi}\xi + \bar{\xi}\chi)$ as the "gauge-fixing term".

We conclude that $\mathcal{L}_{12}$ describes a fermion field transforming in the spin-one representation of the rotation group, with some features reminiscent of a gauge field, namely that in the UV limit the only remaining degrees of freedom are transverse, i.e., helicity eigenstates, so that six components are reduced to four. The Noether currents corresponding to symmetries (42) and (43) are given by

$$j_\mu, j_{A\mu} = -\frac{i}{4}\left[\bar{\chi}\lambda_\mu\chi \mp \bar{\xi}\lambda_\mu\xi\right]. \tag{74}$$

Assigning 3 as the timelike direction, we identify a fermion charge operator $-i\lambda^3 \otimes \sigma_3$ with $\pm$ restframe eigenstates $F, \bar{F} = (1, \mp i, 0, 1, \pm i, 0)^T$, i.e., fermions (antifermions) correspond to left(right)- and right(left)-handed circularly polarised $\chi(\xi)$ states, which remain transverse under SO(3) rotations. Asymptotically, the propagator scales as $k^{-1}$, as expected for a relativistic fermion. The propagator pole at $k^2 = -m^2$ again corresponds to a physical particle.

Finally, we can use the fermion current of (74) to write the Lagrangian for the Thirring model based on $\mathcal{L}_{12}$, using the Y-basis:

$$\mathcal{L}_{\text{rThir}} = \frac{1}{4}\bar{Y}(\partial\cdot\lambda\otimes\sigma_3 + m\mathbb{1}\otimes\sigma_1)Y + \frac{g^2}{32}\left(\bar{Y}\lambda_\mu\otimes\sigma_3 Y\right)^2. \tag{75}$$

In the same basis, the invariances (42) and (43) read

$$Y \mapsto e^{i\alpha}Y \quad ; \quad \bar{Y} \mapsto \bar{Y}e^{-i\alpha} \tag{76}$$

$$Y \mapsto e^{i(\mathbb{1}\otimes\sigma_3)\alpha}Y \quad ; \quad \bar{Y} \mapsto \bar{Y}e^{-i(\mathbb{1}\otimes\sigma_3)\alpha} \tag{77}$$

while parity is

$$Y(x) \mapsto -(\mathbb{1}\otimes\sigma_3)Y(-x); \quad \bar{Y}(x) \mapsto \bar{Y}(-x)(\mathbb{1}\otimes\sigma_3). \tag{78}$$

## 8. Discussion

This paper has developed the description of relativistic fermions in the language of differential geometry, originally set out in [12], to three spacetime dimensions. The principal result is the specification of a continuum field theory sharing the same parity and global U(N)⊗U(N) invariances as the "staggered Thirring model" originally studied numerically using lattice field theory simulations in [6]. In our view, this puts the staggered Thirring

model on a firm footing as an interacting quantum field theory distinct from the U(2N)-invariant version based on the action (1), which is the focus of much recent numerical work [1]. This result is entirely consistent with Becher and Joos' demonstration that Kähler-Dirac fermions are the correct continuum limit for staggered lattice fermions [11]. Beyond the weak-coupling and long-wavelength limits, we have seen that spin/taste entanglement is not merely a lattice artifact, but a genuine feature of an interacting continuum field theory: tastes are not the same as flavours.

An important consequence of regarding the $\varphi$-basis as more fundamental than the more familiar $\psi$-basis is the response to quantum corrections encapsulated in the proposed relation (54) relating renormalised to bare fields, in which multiplicative renormlisation depends solely on $p$, consistent with U(N)⊗U(N) symmetry. This was demonstrated explicitly in Section 6 through the recovery of interaction currents entangling spin and taste originally found in the staggered Thirring model. However, a more spectacular, if speculative, consequence was worked out in Section 7, where the assumption of a strong hierarchy of the $Z_p$ arising due to large anomalous scaling dimensions in the vicinity of a renormalisation-group fixed point motivated the investigation of truncated actions retaining just two $p$-values. The Lagrangian $\mathcal{L}_{\text{rThir}}$ (75) is especially interesting, describing six-component spin-one fermions, with fermions/antifermions being states of well-defined polarisation, and dynamics dominated by the four components lying in the transverse subspace in the UV limit. Could these exotica form the basis for a description of strongly interacting fixed-point dynamics? The answer must await a controlled non-perturbative investigation.

We conclude with a brief discussion of spin and statistics. The Lagrangian (75) describes spin-one fermions which in the canonical approach to field quantisation would be represented by field operators with the anticommutator $\{Y_\alpha(\vec{x},t), Y_\beta^\dagger(\vec{x}',t)\} = \delta^2(\vec{x}-\vec{x}')\delta_{\alpha\beta}$. An immediate concern is the apparent contradiction with the spin-statistics theorem requiring Lorentz-invariant theories of anticommuting fields to be quantised with half-integer spin representations of the Lorentz group. A symptom of the problem is revealed through the ground state expectation of the anticommutator of fields at arbitrary spacetime separation [18]:

$$\langle 0|\{Y(x), \bar{Y}(x')\}|0\rangle = i(i\partial \cdot \tilde{\lambda} \otimes \sigma_3 + m\mathbb{1} \otimes \sigma_1)\Delta_{\text{sym}}(x'-x). \tag{79}$$

Here, $\tilde{\lambda}_\mu$ represents Minkowski space versions of the $\lambda$-matrices, we have assumed that all states are defined in the transverse subspace, and for field quantisation with the "wrong" statistics, the *PCT* theorem dictates the appearance on the RHS of the symmetric solution of the Klein–Gordon equation (or its generalisation):

$$\Delta_{\text{sym}}(x) = \int \frac{d^2\vec{k}}{(2\pi)^2} \frac{\cos(k \cdot x)}{\omega(k)}, \tag{80}$$

where for free fields $\omega(k) = \sqrt{k^2+m^2}$. Specialising to the case of a spacelike interval $\vec{x} = x\hat{x}$ with $|\hat{x}| = 1$, we find

$$\langle 0|\{Y(0), \bar{Y}(\vec{x})\}|0\rangle = \frac{im^{\frac{3}{2}}}{(2\pi x)^{\frac{1}{2}}} \frac{1}{\pi} \left[(i\hat{x} \cdot \tilde{\lambda} \otimes \sigma_3)K_{\frac{3}{2}}(mx) + (\mathbb{1} \otimes \sigma_1)K_{\frac{1}{2}}(mx)\right]. \tag{81}$$

Since the RHS of (79) does not vanish outside the lightcone, there is a violation of microcausality. This is a general result independent of the detailed form of the dispersion $\omega(k)$. For free fields, the asymptotic properties of the modified Bessel functions in (81) can be used to to find

$$\lim_{x\to\infty} \langle 0|\{Y(0), \bar{Y}(\vec{x})\}|0\rangle = \frac{i}{2\pi} \frac{m}{x} e^{-mx} [i\hat{x} \cdot \tilde{\lambda} \otimes \sigma_3 + \mathbb{1} \otimes \sigma_1], \tag{82}$$

and

$$\lim_{x \to 0} \langle 0 | \{ Y(0), \bar{Y}(\vec{x}) \} | 0 \rangle = \frac{i}{2\pi x^2} (i\hat{x} \cdot \tilde{\lambda} \otimes \sigma_3) + \frac{im}{2\pi x} (\mathbb{1} \otimes \sigma_1); \tag{83}$$

that is, the causality violation is localised to within roughly a Compton wavelength of the lightcone, but diverges as $x \to 0$, although less severely than the $x^{-3}$ behaviour of 3 + 1$d$ [18].

Since microcausality is a desirable property for a fundamental theory, the correct relation between spin and statistics is a necessary ingredient of a complete quantum field theory. By hypothesis, however, the spin-one action (75) serves only as an effective description of the dynamics near a UV fixed point, in the deep Euclidean regime $k^2 \to \infty$ very far from the lightcone. The question of whether the spin-statistics linkage compromises the fixed-point description remains open.

**Funding:** This research was funded by STFC Consolidated Grant ST/T000813/1.

**Data Availability Statement:** Not applicable.

**Conflicts of Interest:** The author declares no conflict of interest.

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
