# Peer review of "The Planar Thirring Model with Kähler-Dirac Fermions"

_symmetry, doi:10.3390/sym13081523_

Round 1

Reviewer 1 Report

The author has applied the kahler geometric approach in three-dimensional space-time to treat relativistic fermion in order to construct the Thirring model. The author discussed various aspects of this model. We think it is well-written paper.  The derivation is very clear.

Author Response

I am very pleased the reviewer finds the paper well-written

Reviewer 2 Report

The aim of this paper is to develop a description of strongly interacting relativistic fermions in  three Euclidean spacetime dimensions based on the Kahler’s  geometrical description of fermion fields. 

The realm of application is the Thirring model, and more specifically, in this paper a version of the Thirring model with contact interactions between conserved Noether currents has been analyzed.

In my opinion, this paper is very well written and with a useful self-contained introduction to the differential geometry tools required to the purpose of this work. 

The results obtained on the interpretation of the continuum limit of staggered lattice fermions and on spin/taste entanglement are very interesting and deserve to be published on Symmetry.

Author Response

I am very pleased the reviewer finds the introduction self-contained, and finds the remarks on spin/taste entanglement interesting. 

Reviewer 3 Report

In this paper, the author has studied "The Planar Thirring Model with Kähler-Dirac Fermions". My comments are appended below:

  1. This paper is written in a very comprehensive fashion and details are helpful for every potential general reader.
  2. The underlying concept of this work is very interesting and presented in a very appropriate way.
  3. The introduction, conclusion, and main text part of the paper is written in such a way that no calculation and no information is in the black box.
  4. I also did not have found any computational and conceptual flaws or errors in the draft.
  5. My only point is regarding the references. It would be suggested to the authors to include some of the recent and fundamental references on this subject which will be helpful further for the readers. I will not explicitly point any reference here. It would be really important and also helpful if the authors can able to so.

In view of the above facts, I ask for a minor revision in the draft and once the references will be properly/appropriately included I will give my final decision regarding the publication of this paper in Symmetry,

Author Response

I am very pleased with the reviewers comments (1)-(4).

Regarding (5), it is of course always possible to improve a manuscript by adding references to others' work, and I am happy to include any the referee considers important. I can honestly say I haven't omitted any useful references that I know of. 

For a background to the theoretical issues surrounding the 2+1d Thirring model at strong coupling I have already referred to a recent review [1] which in some sense complements (and indeed stimulates) the current paper. For recent developments in the application of Kaehler-Dirac fermions I have added a 2018 reference [14] outlining their application to dynamical triangulation approaches to quantum gravity, which appears to me to be a very promising theoretical direction. I am unaware of any other important recent paper in this field.

Reviewer 4 Report

The paper is an important contribution to our understanding of strongly interacting fermions in 2+1 dimensions. It provides a solid theoretical ground for the effective continuum description of interacting staggered fermions in 2 + 1 dimensions in terms of differential forms. While a similar description was known for 3+1-dimensional staggered fermions, to my knowledge it has never been worked out for 2+1-dimensional systems, where an additional symmetry appears alongside the usual chiral rotations generated by gamma5, the fifth Dirac matrix. In particular, the paper offers a new perspective on renormalization-group analysis of interacting fermions in 2+1 dimensions. It will certainly be a useful reference for many numerical studies of interacting fermions using staggered fermions on the lattice. I recommend the paper for publication.

Author Response

I am very pleased with the positive comments, in particular noting the novel application of the KD approach to 2+1 dimensions.